# HUMANVIDEO-MME: BENCHMARKING MLLMS FOR HUMAN-CENTRIC VIDEO UNDERSTANDING

## ABSTRACT

Multimodal Large Language Models (MLLMs) have demonstrated significant advances in visual understanding tasks involving both images and videos. However, their capacity to comprehend human-centric video data remains underexplored, primarily due to the absence of comprehensive and high-quality evaluation benchmarks. Existing human-centric benchmarks predominantly emphasize video generation quality and action recognition, while *overlooking essential perceptual and cognitive abilities* required in human-centered scenarios. Furthermore, they are often limited by single-question paradigms and overly simplistic evaluation metrics. To address above limitations, we propose a modern **HumanVideo-MME**, a rigorously curated benchmark designed to provide a more holistic evaluation of MLLMs in human-centric video understanding. Compared to existing human-centric video benchmarks, our work offers the following key features: **(1) Diverse evaluation dimensions**: HumanVideo-MME encompasses 13 tasks, ranging from basic attribute perception (*e.g.*, age estimation, emotion recognition) to advanced cognitive reasoning (e.g., social relationship prediction, intention prediction), enabling comprehensive assessment of model capabilities; **(2) Varied data types**: The benchmark includes multiple-choice, fill-in-blank, true/false, and open-ended question formats, combined with diverse evaluation metrics, to more accurately and robustly reflect model performance; **(3) Multi-domain video coverage**: The benchmark spans 50 distinct visual scenarios, enabling comprehensive evaluation across fine-grained scene variations; **(4) Temporal coverage**: The benchmark covers videos from short-term (10 seconds) to long-term (up to 30min) durations, supporting systematic analysis of models' temporal reasoning abilities across diverse contextual lengths. We evaluate several advanced open-source MLLMs on the HumanVideo-MME. While models excel in closed-form tasks, their performance drops sharply in open-ended generation, revealing a reliance on shallow patterns over genuine reasoning. In contrast, fill-in-blank and open-ended formats better expose reasoning challenges in human behavior understanding. By spanning diverse tasks and paradigms, HumanVideo-MME systematically reveals these limitations and facilitates the MLLM development.

## 1 INTRODUCTION

Recent advances in Multimodal Large Language Models (MLLMs) (Hurst et al., 2024; Liu et al., 2024b; Chen et al., 2024b; Bai et al., 2025; Team et al., 2024) have demonstrated remarkable capabilities in perceptual understanding and reasoning for general video comprehension tasks. Among various types of video data, human-centric videos represent a particularly critical domain due to their prevalence in real-world data. Compared to general video understanding, human-centric video understanding imposes greater challenges on models, as these tasks require not only the recognition of human actions and behaviors but also more sophisticated reasoning abilities. A systematic investigation of current MLLMs' capabilities and limitations in this domain is therefore critical for advancing both theoretical frameworks and practical applications. However, existing benchmarks suffer from three fundamental limitations: *1)* overly simplistic evaluation dimensions that fail to comprehensively cover the wide range of human-centric tasks; *2)* restricted question-answer paradigms that overlook more complex and diverse reasoning needs; and *3)* limited temporal coverage and scenario diversity, which hinder evaluation of MLLMs' generalization capabilities.

These shortcomings inevitably undermine a comprehensive assessment of MLLMs' true potential in human-centric video understanding.

To bridge this gap, we propose HumanVideo-MME , a human-centered video understanding evaluation benchmark. Compared to existing benchmarks (Li et al., 2024b; Zhou et al., 2024), our benchmark stands out in three key innovations: *1)* **Diverse Evaluation Dimensions**: Covering 13 cognitive tasks, it systematically evaluates MLLMs' capabilities in both basic perception, such as age/gender recognition, emotion detection, action identification, and higher-order cognition, such as behavioral intention prediction and social relationship inference; *2)* **Novel Evaluation Paradigms**: We propose a multi-paradigm framework that integrates various interaction modes-such as multiple-choice, fill-in-blank, and open-ended question answering—along with comprehensive quantitative metrics to provide holistic, application-oriented assessments; *3)* **Spatiotemporal Coverage**: Encompassing a wide range of 50 specidomains, with video durations ranging from 10 seconds to 30 minutes, this benchmark effectively evaluates models' ability to capture complex spatiotemporal relationships.

To evaluate the effectiveness of HumanVideo-MME , we conduct extensive benchmarking across several state-of-the-art open-source MLLMs (Chen et al., 2024b; Bai et al., 2025; Li et al., 2024a; Cheng et al., 2024). Our results reveal that while several models achieve strong performance in Multiple-Choice (MC) and True/False (TF) formats, their performance notably degrades in Fill-In-Blank (FIB) and Open-Ended Questions (OEQ) formats. For instance, while Qwen2.5-VL-32B (Bai et al., 2025) achieves an impressive 94.77% accuracy on multiple-choice causal reasoning tasks, its F1@1 score in open-ended causal generation drops close to zero (see Tab. 4). This stark contrast suggests that current MLLMs tend to rely heavily on superficial patterns or pretrained priors when solving closed-form questions, rather than engaging in genuine structural reasoning. Furthermore, the models exhibit consistently low accuracy on tasks requiring fine-grained visual perception, such as face recognition, underscoring persistent limitations in capturing subtle identity-related cues, which may be attribute to the scarcity of celebrity-focused data during pre-training. These findings reveal two critical bottlenecks in the existing open-source MLLMs: weak generalization in generative tasks and insufficient grounding in fine-grained perception. By incorporating a diverse set of task types and question formats, HumanVideo-MME  systematically uncovers these limitations, and establishes a rigorous evaluation benchmark to guide the development of future MLLMs.

In summary, our contributions are as follow:

- We construct HumanVideo-MME , a large-scale benchmark tailored for human-centric video understanding. It covers 13 diverse tasks across perception and cognition, supports four QA paradigms (MC, FIB, TF, OEQ), and spans over more than 50 real-world scenarios, enabling spatiotemporal reasoning at multiple granularities.

- We introduce a novel composite evaluation metric for the causal reasoning task that integrates lexical accuracy, structural consistency, and LLMs-based semantic coherence scoring, enabling a more holistic and fine-grained assessment of generative reasoning capabilities.

- Comprehensive experiments reveal a stark contrast between performance in closed-form tasks and generative tasks. Our benchmark effectively exposes these limitations and guide future MLLMs toward better human-centric reasoning.

## 2 RELATED WORK

### 2.1 MULTIMODAL LARGE LANGUAGE MODELS (MLLMS)

The advent of large language models (LLMs) (Touvron et al., 2023; Achiam et al., 2023) has catalyzed significant breakthroughs in MLLMs (Bai et al., 2025; Chen et al., 2024b; Hurst et al., 2024; Cai et al., 2024; Liu et al., 2024a; Team et al., 2024; Ye et al., 2024; Li et al., 2023a) for visual-language understanding. In image-based tasks, MLLMs typically project visual features into the language space and leverage language models to fuse and reason over multimodal information. In contrast, video understanding poses greater challenges due to the intrinsic complexity of temporal modeling, demanding more sophisticated mechanisms to capture both spatial and temporal dependencies. Recent efforts have led to the development of several Video MLLMs (Li et al., 2023b; Maaz et al., 2024; Lin et al., 2024; Xu et al., 2024) aiming to improve the understanding of temporal visual information. For example, Video-LLaVA (Lin et al., 2024) introduces a pre-alignment mechanism between image

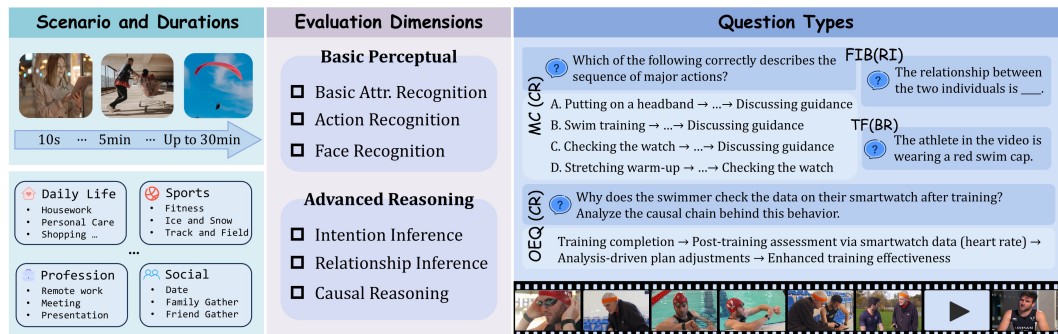

Figure 1: **Overview of HumanVideo-MME** that spans diverse human-centric scenarios (50+ domains in 10s~30mins) and covers both basic perception and advanced reasoning tasks. It supports Multiple-Choice (MC), Fill-in-Blank (FIB), True/False (TF), and Open-Ended Questions (OEQ) to comprehensively evaluate MLLMs' understanding and cognitive capabilities.

and video features for multimodal instruction tuning, thereby achieving unified modeling across visual modalities. VideoChat (Li et al., 2023b) utilizes video foundation models to encode videos as embeddings, and feeds them directly into an LLM, enabling end-to-end video question answering.

Despite these advancements in architecture, the potential of current video MLLMs for human-centric video understanding remains largely underexplored. To address this gap, we propose a dedicated benchmark focusing on human-centric video comprehension, which systematically evaluates model performance across both basic perceptual tasks and higher-order cognitive reasoning challenges.

## 2.2 MLLM Benchmarks

In the field of MLLMs, numerous benchmarks have been developed to evaluate models' capabilities in both perception and cognition. As image-based MLLMs have demonstrated impressive performance across various benchmarks (Goyal et al., 2017; Gurari et al., 2018; Hudson and Manning, 2019; Lu et al., 2022; Liu et al., 2023; Yue et al., 2024; Fu et al., 2023; Li et al., 2023c), research attention has increasingly shifted toward video understanding, an inherently more complex domain, thus driving the need for high-quality video benchmarks. Existing video datasets (Fu et al., 2024; Mangalam et al., 2023; Li et al., 2024c; Ning et al., 2023) primarily focus on general video understanding tasks, such as object detection tasks. However, human-centric videos, which dominate real-world video content, present significantly greater challenges for multimodal understanding. Unlike general video tasks, understanding human-centric videos often requires comprehensive reasoning about human behaviors, intentions, emotional states, and social relationships.

Current human-centric video benchmarks can be categorized into two types: video generation evaluation (Yu et al., 2019; Li et al., 2024b; Wang et al., 2024b; Huang et al., 2024) and understanding assessment (Zhou et al., 2024). HumanVid (Wang et al., 2024b) combines real and high-quality synthetic videos with precisely annotated human poses to support research in portrait animation. OpenHumanVid (Li et al., 2024b) integrates structured text, skeletal motion data, and speech to improve consistency and semantic alignment in video generation. In the domain of video understanding HumanVBench (Zhou et al., 2024) has recently been proposed to focus on inner emotions and their external manifestations, introducing several fine-grained tasks such as emotion recognition. However, existing benchmarks suffer from oversimplified evaluation dimensions, restricted questioning paradigms, and limited temporal/scenario coverage. These shortcomings inevitably hinder a comprehensive assessment of the true potential of MLLMs in human-centric video understanding.

## 3 HumanVideo-MME

HumanVideo-MME dataset construction process involves three primary steps as shown in Fig. 2: video collection, automated question-answer annotation, and quality review.

### 3.1 Video Source and Collection

Our video data is sourced from publicly available datasets, including UltraVideo (Xue et al., 2025) and OpenHumanVid (Li et al., 2024b), as well as Koala-36M (Wang et al., 2024a). To ensure that

Figure 2: **HumanVideo-MME construction pipeline.** The benchmark is built through a three-stage pipeline: (1) large-scale Video Collection across diverse human-centric domains; (2) Automated QA annotation via MLLMs and structured templates; (3) a two-tier Quality Review combining automatic filtering and expert verification to ensure annotation reliability.

the dataset comprehensively covers a wide variety of human-centric contexts, we first define seven core domains to collect diverse data. These domains include: daily life, professional activities, social interactions, health and medical management, education and learning, transportation, and cultural entertainment. Each domain is further subdivided into specific scenarios to capture finer-grained contextual information. For instance, within the entertainment domain, we focus on scenes such as street performances. Ultimately, the HumanVideo-MME includes 50 meticulously categorized human-centered scenarios, and the full scene classification is shown in Fig. 3(a).

Sepcifically, data collection is conducted by 10 qualified researchers following strict quality control protocols. Videos were required to *1)* align with predefined scenario classifications, *2)* exclude artificial elements such as subtitles and black borders, and *3)* maintain resolution above 720P. The final dataset comprises 1160 videos, with durations ranging from 10 seconds to 30 minutes, ensuring a diversity that spans from short-term interactions to longer-duration events.

## 3.2 AUTOMATED QUESTION-ANSWER ANNOTATION PIPELINE

We meticulously design an automated annotation pipeline that consists of two key steps: *1)* Attribute Labeling determines which types of evaluation tasks are suitable for constructing questions based on the video content. *2)* Question-Answer Generation leverages the state-of-the-art MLLMs to automatically generate question-answer pairs for the relevant attributes, while the distractor generation further refines distractors, ensuring that they are not only distinct from the correct answers but also sufficiently challenging to enhance the evaluation's rigor.

**Video Captioning and Attribute Labeling.** We categorize the evaluation dimensions of MLLMs into basic perception and advanced reasoning tasks. Basic perception tasks include human attribute recognition, face detection, and action recognition, while advanced reasoning tasks involve relationship inference, intent and motivation inference, and causal reasoning. To ensure that the generated questions are more targeted, we first perform Attribute Labeling on the video to identify the most relevant evaluation tasks.

To ensure high-quality attribute labeling grounded in both visual and semantic understanding, we adopt a two-stage model pipeline. Specifically, we employ the state-of-the-art Qwen2.5-VL-72B (Bai et al., 2025), to generate detailed video captions that encapsulate essential scene elements, human actions, and contextual cues from raw video inputs. These captions serve as a rich semantic representation of the visual content. Subsequently, we utilize the Qwen2.5-72B (Yang et al., 2024), which excels at language-based reasoning, to infer the most appropriate task-specific attribute labels based on the generated captions.

This decoupled framework which separates visual captioning from language-based label inference offers several key advantages. First, by embedding attribute labels within natural language descriptions, it enhances interpretability, making the allocation of evaluation dimensions more transparent and easier to understand. Second, it enables each model component to operate within its area of expertise, thereby mitigating error propagation: the MLLM focuses on extracting rich, structured semantic content from video, while the LLM performs high-level reasoning and task assignment based on textual cues. Overall, this approach bridges low-level perceptual signals and high-level reasoning objectives, enabling the generation of more coherent and targeted queries for complex, human-centric video understanding tasks.

**Question-Answer Generation.** To ensure the diversity and accuracy of video question–answer pairs, we manually designed 5∼10 varied question templates for each secondary evaluation task (*e.g.*, action recognition in basic attribute perception, causal reasoning in advanced reasoning inference). Prior to generating QA pairs, we first activate the corresponding attributes based on the annotations during the attribute labeling stage: only if a particular attribute is marked do we trigger the related question generation, thereby avoiding irrelevant or low-quality pairs. During template-based question generation, we dynamically insert localization cues, such as "the pedestrian on the far left" or "the man wearing a blue shirt", according to scene complexity to eliminate ambiguity in multi-person contexts. Next, we invoke the Qwen2.5-VL-72B (Bai et al., 2025) to simultaneously produce the single correct answer and at least three distractors closely tied to key visual information; distractor design strategies include feature-detail variations (e.g., slight color or count adjustments) and temporal displacements (e.g., action-sequence perturbations) to heighten cognitive challenge. Through this pipeline, we ensure that the generated QA pairs align tightly with the original video content while balancing evaluation depth and difficulty.

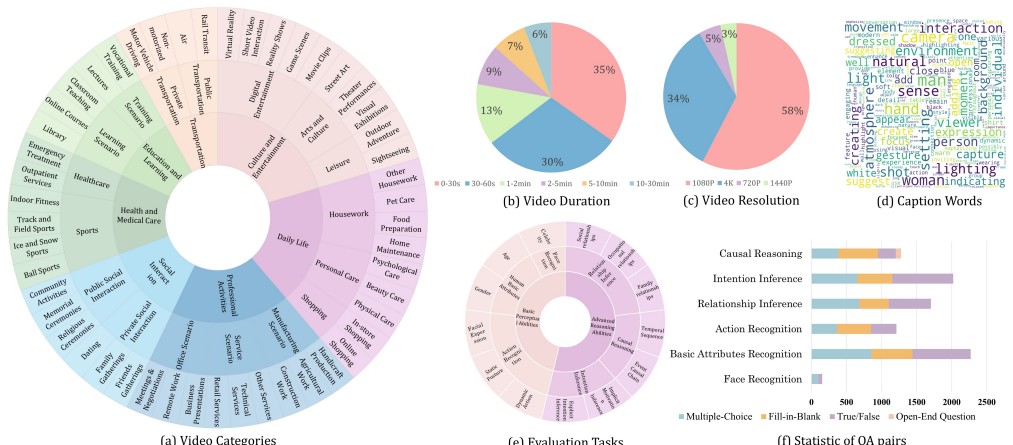

Figure 3: **Statistics of HumanVideo-MME** . (a) Our benchmark covers 50+ human-centric categories across diverse domains. (b) Video durations range from short clips to long-form content. (c) Most videos are in high resolution (720P or above), supporting fine-grained visual analysis. (d) The caption vocabulary covers diverse semantic cues, emphasizing human appearance, actions, and interactions. (e) Our evaluation tasks span both perceptual and high-level reasoning abilities. (f) The statistic information of QA pairs.

## 3.3 MANUAL QUALITY REVIEW AND ADJUSTMENT

To ensure the reliability and diversity of the dataset, we designed a rigorous review process consisting of following two main steps.

**Automated Review.** Inspried by previous successes, we provide pure-text questions to the state-of-the-art LLMs like Qwen2.5 (Yang et al., 2024) and filter out low-quality questions where the answer is implicitly provided in the question or does not rely on understanding the video content. For example, questions like "What is the gender of the male character wearing a blue T-shirt in the video?" will not effectively evaluate the model's comprehension ability and are discarded in this step. The remaining questions are then subjected to further processing in the manual review phase.

**Manual Review.** We employ a two-stage independent human review combined with majority voting to ensure the quality of the dataset. For MC and T/F questions, two experienced reviewers independently vote on each item, casting their votes and suggesting alternative options for any unreasonable choices. When both reviewers reach consensus on the validity of the original answer or select the same alternative, the QA pair is accepted. In cases of disagreement, a third senior expert adjudicates the final decision. For fill-in-blank and open-ended questions, annotators are required to conduct a thorough assessment of the validity of all candidate answers. A question-answer pair is preserved only if both rounds of review confirm the reasonableness of all options. In summary, our review pipeline balances the efficiency of automated filtering with the rigor of manual inspection, substantially reducing labor costs while ensuring the accuracy of the question-answer pairs.

## 3.4 DATASET STATISTICS AND ANALYSIS

**Video Statistics.** Our benchmark dataset comprises 1,200 videos spanning multiple domains (Fig. 3(a)), including social interactions, daily life scenarios, professional activities, sports, medical procedures, educational content, and entertainment. This comprehensive categorization captures the full spectrum of real-world human activities. Fig. 3(b) illustrates the temporal distribution of video durations, which range from brief clips (<30 seconds) to extensive segments (up to 30 minutes). This diversity in duration enables comprehensive evaluation of Video MLLMs across varying temporal dynamics, from instantaneous action recognition to complex sequence analysis requiring extended temporal comprehension. Fig. 3(c) presents the distribution of video resolutions, which range from 720P to 4K and above. Notably, the majority of videos have resolutions exceeding 1080P, underscoring the high quality of our dataset.

Table 1: **Comparison of prominent benchmarks in the video domain.** We highlight key attributes, including domain scope (Open *vs.* Human), the number of videos (#Videos), the number of question-answer instances (#QA Ins.), supported tasks, QA formats, evaluation metrics and resolution (Res). MC, FIB, TF, and OEQ are the abbreviation of multiple-choice, fill-in-the-blank, true/false, and open-ended questions, respectively.

| Datasets | Year | Domain | #Videos | #QA Ins. | Tasks | QA formats | Metrics | Res |
|---|---|---|---|---|---|---|---|---|
| Video-MME (Fu et al., 2024) | 2024.05 | Open | 900 | 2700 | Understanding | MC | Acc | / |
| OpenVid-1M (Nan et al., 2025) | 2024.07 | Open | 1M | / | Synthetic | / | / | / |
| Panda-70M (Chen et al., 2024a) | 2024.02 | Open | 70M | / | Synthetic | / | / | 720P |
| Koala-36M (Wang et al., 2024a) | 2024.10 | Open | 36M | / | Synthetic | / | / | 720P |
| Tiktok-V4 (Chang et al., 2023) | 2023.11 | Dance | 350 | / | / | / | / | / |
| OpenHumanVid (Li et al., 2024b) | 2024.12 | Human | 13.2M | / | Synthetic | / | / | 720P |
| HumanVBench (Zhou et al., 2024) | 2024.12 | Human | / | 2116 | Understanding | MC | Acc. | / |
| Ours | 2025.05 | Human | 1200 | 8700 | Understanding | MC, FIB, TF, OEQ | Acc., F1, LLMs | 720P-4K |

**QA Statistics.** Under our proposed HumanVideo-MME , the designed task types encompass multiple QA formats, enabling comprehensive evaluation from basic perception to high-level cognition. Specifically, the Basic Attribute Recognition task is the largest, comprising 2,517 QA pairs across MC, FIB, and TF formats, reflecting broad coverage of fundamental human attribute perception. The Intention Inference and Relationship Inference tasks include 1,938 and 1,399 QA pairs, respectively—ranking second only to Basic Attribute Recognition—demonstrating our method's strong coverage in advanced cognitive tasks. The Causal Reasoning task is the only evaluation involving open-ended questions, consisting of 70 OEQ samples (approximately 7.5% of this task's total), designed to evaluate the model's high-level reasoning abilities beyond fixed-answer paradigms. In addition, although Action Recognition and Face Recognition involve fewer samples, they serve as essential components at the perceptual level, contributing fine-grained understanding of human actions and identity, and thereby complementing the overall cognitive evaluation.

**Comparison with Related Benchmarks.** Table 1 compares our proposed HumanVideo-MME with several existing benchmarks across different attributes, highlighting its unique advantages in human-centric multimodal video understanding. *1) Comparison with open-domain benchmarks.* These benchmarks, such as OpenVid-1M (Nan et al., 2025) and Koala-36M (Wang et al., 2024a), primarily focus on diverse video content from various domains. These datasets are large-scale, encompassing extensive video collections; for example, Panda contains over 70 million videos. Such benchmarks typically emphasize general video understanding tasks, including object detection and other perception-oriented objectives. However, they do not address the complexities inherent in human-centric video tasks, which require models to interpret human actions, emotions, intentions, and social interactions. While these open-domain benchmarks are valuable for evaluating general video understanding, their lack of focus on human-centric aspects limits their ability to assess models in scenarios involving complex human behaviors and higher-order cognitive reasoning.

*2) Comparison with Human-centric benchmarks.* Unlike benchmarks such as HumanVid (Wang et al., 2024b) and OpenHumanVid (Li et al., 2024b), which primarily focus on evaluating video synthesis tasks, both HumanVBench (Zhou et al., 2024) and our proposed HumanVideo-MME are designed to assess video understanding tasks. Compared to HumanVBench, HumanVideo-MME encompasses a broader range of cognitive tasks, including higher-order reasoning tasks such as relationship inference, intention prediction, and causal reasoning. This diversity makes HumanVideo-MME a more robust benchmark for evaluating human-centric video. Another key distinction lies in temporal coverage: HumanVideo-MME accommodates video segments ranging

from brief clips (approximately 10s) to extended sequences (up to 30 minutes), significantly surpassing HumanVBench's predominant focus on sub-10-second content. Furthermore, HumanVideo-MME employs a variety of question formats, including multiple choice (MC), fill-in-blank (FIB), true/false (TF), and open-ended questions (OEQ), as well as diverse evaluation metrics such as accuracy and LLM-based evaluation. This comprehensive approach enables a more thorough evaluation of model performance compared to benchmarks with a single question type or metric.

# 4 BENCHMARKING MLLMs IN HUMANVIDEO-MME

## 4.1 EXPERIMENTAL SETTING

### 4.1.1 EVALUATION DETAILS

**Evaluation MLLMs.** To comprehensively evaluate the understanding capabilities of open-source MLLMs in human-centric video scenarios, we benchmarked several state-of-the-art models, including Qwen2.5-VL series (Bai et al., 2025), InternVL2.5 series (Chen et al., 2024b), LLaVA-OneVision (Li et al., 2024a), LLaVA-Video (Zhang et al., 2024), VideoLLaMA2 (Cheng et al., 2024).

**Implementation Details.** For MC and TF questions, we adopt the prompt: "Answer with the option's letter from the given choices directly and only give the best option." This ensures that the model outputs a single, unambiguous letter corresponding to its top choice. For FIB tasks, we append the instruction "Answer in short" to encourage concise and direct responses. For open-ended questions (OEQ), we guide the model with the prompt Use "$\rightarrow$" to connect cause and effect explanations" to promote structured and interpretable outputs. All evaluations are conducted on 8 NVIDIA H20 GPUs. The batch size is set to 1.

### 4.1.2 METRICS

**MC and TF.** Following prior work in MLLMs, we adopt the standard Accuracy metric to evaluate model performance on MC and TF.

**FIB.** To comprehensively evaluate the model's ability to generate accurate and semantically appropriate answers in the cloze-style QA setting, we employ Top-1 evaluation strategies. Each question may correspond to multiple acceptable answers, and the ground-truth label for each sample is formulated as a set of valid candidate responses. The model is evaluated based on whether its prediction matches any of these candidates. Under the Top-1 setting, the model is required to produce a single answer per sample. We adopt the following standard metrics: *Precision (Precision@1), Recall (Recall@1) and F1 (F1@1):* Precision@1 is defined as the proportion of predicted items that are correct, while Recall@1 measures the proportion of ground-truth answers that are covered. F1@1 is adopted to provide a balanced measure of both correctness and completeness.

**OEQ.** To evaluate the causal reasoning capability of MLLMs, we adopt a hybrid evaluation framework that jointly considers factual accuracy, structural coherence, and semantic plausibility of generated causal chains. Each model prediction is compared against a human-annotated gold causal chain, typically represented as a sequence of events (e.g., "A $\rightarrow$ B $\rightarrow$ C").

(1) Fuzzy Step-wise F1 Score ($Score_F$): we first compute a step-level F1 score that measures the overlap between predicted and ground-truth causal events using fuzzy token-based matching. This score accounts for lexical variation and reflects the factual correctness of the events comprising the chain. (2) To evaluate structural consistency, we compute the Longest Common Subsequence (LCS) between the predicted ($P$) and reference chains ($G$). The final score is normalized by the length of the gold chain: $Score_O = \frac{\text{LCS}(P,G)}{|G|}$. (3) To evaluate the overall causal plausibility beyond step-level matching, we employ the state-of-the-art Qwen2.5-72B as a judge to rate the semantic coherence of each generated chain with respect to the reference. The model assigns a score ($Score_G$) from 0 to 5, where $Score_G = 0$ indicates no clear causal link between the predicted and reference chains, and $Score_G = 5$ represents perfect semantic alignment and logically sound event sequencing. We define a final composite score as a weighted sum of the three components:

$$Score = \alpha \cdot Score_F + \beta \cdot Score_O + \gamma \cdot Score_G^{\text{norm}}, \tag{1}$$

where $Score_G^{\text{norm}}$ is the normalized $Score_G$ ($Score_G^{\text{norm}} = \frac{Score_G}{5}$), and the default weights are $\alpha = 0.5$, $\beta = 0.3$, and $\gamma = 0.5$.

Table 2: Performance of different open-sourced MLLMs on HumanVideo-MME under the Multiple-Choice and True/False questions, respectively. BR, FR, AR are the abbreviate of Basic Attribute Recognition, Action Recognition, and Face Recognition. RI, InI, CR are the abbreviate of Relation Inference, Intention Inference, and Causal Reasoning, respectively.

| Model | Multiple-Choice | | | | | | | True/False | | | | | | |
|---|---|---|---|---|---|---|---|---|---|---|---|---|---|---|
| | BR | FR | AR | RI | InI | CR | Avg | BR | FR | AR | RI | InI | CR | Avg |
| VideoLLaMA2-7B | 75.73 | 84.84 | 88.56 | 72.14 | 88.52 | 84.59 | 82.40 | 73.10 | 81.15 | 69.98 | 88.64 | 93.66 | 76.12 | 80.44 |
| LLaVAVideo-7B | 87.85 | 65.59 | 97.46 | 95.86 | 94.73 | 91.13 | 88.77 | 71.90 | 75.94 | 75.23 | 90.41 | 92.30 | 82.18 | 81.33 |
| LLaVAOneVision-7B | 87.01 | 85.30 | 95.61 | 95.38 | 95.45 | 92.07 | 91.80 | 73.68 | 84.90 | 79.05 | 91.54 | 95.60 | 83.33 | 84.68 |
| Qwen2-VL-7B | 85.55 | 42.83 | 93.53 | 95.74 | 96.60 | 95.66 | 84.98 | 80.78 | 79.68 | 77.68 | 90.33 | 96.24 | 79.36 | 84.01 |
| Qwen2.5-VL-7B | 87.00 | 61.19 | 94.51 | 94.03 | 95.81 | 93.87 | 87.73 | 90.42 | 87.17 | 82.08 | 91.95 | 93.13 | 84.85 | 88.60 |
| Intern2.5-VL-8B | 83.28 | 73.73 | 91.14 | 85.55 | 91.22 | 84.93 | 84.98 | 84.60 | 90.91 | 74.05 | 79.35 | 85.66 | 75.32 | 81.65 |
| Qwen2.5-VL-32B | 88.70 | 49.77 | 96.90 | 95.91 | 96.47 | 94.77 | 87.09 | 88.09 | 93.99 | 83.13 | 92.04 | 96.11 | 85.77 | 89.86 |

## 4.2 MAIN RESULTS

### 4.2.1 EVALUATION UNDER THE MC AND TF QUESTION FORMATS

Table 2 reports the performance of recent state-of-the-art MLLMs on HumanVideo-MME under both Multiple-Choice (MC) and True/False (TF) formats. In the MC setting, models tend to perform well on high-level tasks such as Intention Inference (InI) and Causal Reasoning (CR), where Qwen2-VL-7B and Qwen2.5-VL-32B reaches the best 96.60% / 95.66% and 96.47% / 94.77% respectively. However, their performance significant drop in Face Recognition (FR) task (42.83% and 49.77%), revealing their limitations in fine-grained facial understanding. At the same time, the scarcity of celebrity-centric samples in the pretraining corpus may also contribute to this degradation. This suggests that although large-scale MLLMs has been effectively equipped MLLMs with the ability to follow causal and relational logic, their capacity for fine-grained facial discrimination, particularly in capturing subtle identity-related cues under scarce celebrity samples, remains a major bottleneck. The performance trends in the True/False (TF) setting are generally consistent with those observed in MC. Compared to perceptual tasks, most models demonstrate better performance on advanced reasoning tasks.

### 4.2.2 EVALUATION UNDER THE FIB QUESTION FORMATS

Table 3 presents the performance of MLLMs on the HumanVideo-MME under the Fill-in-Blank (FIB) question format. In perception-level tasks, models generally achieve higher performance in basic attribute recognition (e.g., age, gender) compared to action recognition. For instance, Qwen2.5-VL-32B and LLaVAVideo-7B achieve Precision@1 scores of 61.2% and 60.2%, respectively, with corresponding F1@1 scores of 13.77% and 13.73%. These results highlight the strength of current MLLMs in low-level semantic understanding. However, in action recognition tasks, despite Qwen2.5-VL-32B achieving the highest Precision@1 (17.77%), its F1@1 score remains low, indicating limited capacity in modeling dynamic behaviors. In contrast, performance substantially degrades in higher-order reasoning tasks. For causal reasoning in particular, nearly all models exhibit F1@1 scores close to zero. The Intern2.5-VL series fails entirely on this task, while LLaVAVideo-7B performs best with an F1@1 of only 0.37. These results underscore the significant limitations of current MLLMs in modeling and generating coherent causal chains.

Notably, this trend contrasts with observations from the multiple-choice (MC) and true/false (TF) question formats. As shown in Table 2, LLaVAOneVision-7B and Qwen2.5-VL-32B achieve MC accuracies of 92.07% and 94.77%, and TF accuracies of 83.33% and 85.77%, respectively, on the causal reasoning task. These findings suggest that in closed-form question settings, models can leverage language priors and pattern memorization from pretraining to match the correct answer, without necessarily performing genuine causal reasoning.

We hypothesize that under the MC or TF formats, models can often eliminate clearly incorrect choices or rely on surface-level keyword matching to select the correct answer. In contrast, the FIB format closely resembles natural language generation tasks, where models must produce responses without explicit options and rely solely on their internal reasoning capabilities. As such, FIB serves as a more challenging and discriminative benchmark for evaluating the true reasoning capacity of MLLMs.

Table 3: Performance of MLLMs on HumanVideo-MME under the fill-in-blank questions. We employ Precision@1, Recall@1, and F1@1 for evaluation, respectively.

| Model | Tasks | | | | |
| --- | --- | --- | --- | --- | --- |
| | Action Recognition | Basic Attribute Recognition | Causal Reasoning | Intention Inference | Relationship Inference |
| VideoLLaMA2-7B | 4.40 / 0.50 / 0.90 | 10.73 / 1.47 / 2.53 | 0.37 / 0.03 / 0.07 | 0.13 / 0.00 / 0.03 | 6.27 / 0.73 / 1.37 |
| LLaVAVideo-7B | 10.57 / 1.37 / 2.33 | 60.23 / 7.90 / 13.73 | 1.93 / 0.20 / 0.37 | 1.53 / 0.13 / 0.30 | 13.37 / 1.87 / 3.13 |
| LLaVAOneVision-7B | 4.27 / 0.50 / 0.87 | 49.37 / 6.50 / 11.30 | 0.00 / 0.00 / 0.00 | 0.77 / 0.07 / 0.13 | 3.77 / 0.43 / 0.83 |
| Qwen2-VL-7B | 13.90 / 1.80 / 2.97 | 34.13 / 4.33 / 7.83 | 1.40 / 0.13 / 0.27 | 3.60 / 0.40 / 0.70 | 11.17 / 1.30 / 2.30 |
| Qwen2.5-VL-7B | 10.80 / 1.33 / 2.30 | 58.03 / 7.50 / 13.10 | 0.00 / 0.00 / 0.00 | 1.20 / 0.13 / 0.23 | 14.53 / 1.97 / 3.37 |
| Intern2.5-VL-8B | 1.13 / 0.13 / 0.27 | 23.50 / 3.13 / 5.47 | 0.00 / 0.00 / 0.00 | 0.27 / 0.03 / 0.07 | 2.93 / 0.33 / 0.63 |
| Qwen2.5-VL-32B | 17.17 / 2.00 / 3.50 | 61.17 / 7.87 / 13.77 | 0.00 / 0.00 / 0.00 | 2.67 / 0.27 / 0.50 | 15.87 / 1.87 / 3.37 |
| Intern2.5-VL-38B | 3.13 / 0.37 / 0.63 | 28.60 / 3.57 / 6.23 | 0.00 / 0.00 / 0.00 | 0.00 / 0.00 / 0.00 | 5.63 / 0.90 / 1.50 |

Table 4: Performance of MLLMs on HumanVideo-MME under the open-ended questions for causal reasoning capacibility evaluation.

| | LLaVA-Video-7B | Qwen2-7B | Intern2.5-VL-8B | Intern2.5-VL-38B | VideoLLaMA2-7B | Qwen2.5-VL-7B | Qwen2.5-VL-32B |
| --- | --- | --- | --- | --- | --- | --- | --- |
| $Score_F$ | 0.14 | 0.15 | 0.15 | 0.17 | 0.19 | 0.22 | 0.19 |
| $Score_O$ | 0.24 | 0.33 | 0.3 | 0.37 | 0.35 | 0.47 | 0.51 |
| $Score_G$ | 0.49 | 0.56 | 0.57 | 0.53 | 0.56 | 0.64 | 0.69 |
| $Score$ | 0.39 | 0.45 | 0.45 | 0.46 | 0.48 | 0.57 | 0.59 |

### 4.2.3 EVALUATION UNDER THE OEQ QUESTION FORMATS

Table 4 presents the ability of various MLLMs to generate coherent causal chains in open-ended question settings. We observe substantial performance differences across models. In terms of $Score_F$, which quantifies the overlap between predicted and reference events using fuzzy token matching, most models perform comparably, ranging between 0.14 and 0.22. This indicates that the lexical accuracy at the event level remains limited. Similarly, the structural consistency score ($Score_O$) reveals that only the strongest models, Qwen2.5-VL-32B (0.51) and Qwen2.5-VL-7B (0.47), are able to retain a meaningful portion of the event sequence. The remaining models exhibit significantly weaker structural alignment.

Semantic plausibility ($Score_G$), assessed by the state-of-the-art LLM through the metric, provides a clearer view of models' higher-level reasoning capabilities. Qwen2.5-VL-32B achieves the highest $Score_G$ (0.69), followed by Qwen2.5-VL-7B (0.64) and Intern2.5-VL-8B (0.57), suggesting that some models are capable of generating causally plausible and semantically coherent chains even when token-level overlaps are limited. In terms of overall performance, Qwen2.5-VL-32B leads with the highest average score (0.59), followed by Qwen2.5-VL-7B (0.57), while all other models score below 0.50. Overall, the joint evaluation across $Score_F$, $Score_O$, and $Score_G$ provides a comprehensive assessment, capturing complementary aspects of model performance in causal reasoning.

## 5 CONCLUSION

In this paper, we presented HumanVideo-MME , a benchmark encompassing 13 human-centric tasks—ranging from basic perception (e.g., age, emotion) to advanced reasoning (e.g., social relationship, intention, causal inference), and four QA formats (MC, FIB, TF, OEQ). By evaluating several state-of-the-art MLLMs, we revealed that existing MLLMs demonstrate relatively strong performance under MC and TF formats, but significantly underperform in generation-based FIB and OEQ tasks, especially in causal reasoning. This suggests that while current MLLMs may appear capable in structured tasks, they often rely on pattern heuristics instead of genuinely reasoning through the underlying causal or logical structure.

**Limitations and Border Impact** We identify two major directions for future research. First, extending the benchmark to support comprehensive evaluation of generative models. Second, developing more fine-grained evaluation metrics, particularly for open-ended tasks—moving beyond accuracy to enable automatic assessment of reasoning coherence, factual grounding, and causal consistency. We believe these efforts are critical for advancing MLLMs toward deeper and more reliable understanding of complex human behaviors.

## ETHICS STATEMENT

We have ensured that our study and dataset construction follow ethical standards, with no direct involvement of human subjects, and no foreseeable risk of harm. Data usage complies with privacy and legal requirements, and we have aimed to mitigate potential biases in annotations and model evaluation. We disclose no conflicts of interest or sponsorship that could influence the results.

## REPRODUCIBILITY STATEMENT

We have already elaborated on all the models or algorithms proposed, experimental configurations, and benchmarks used in the experiments in the main body or appendix of this paper. Furthermore, we declare that the entire code used in this work will be released after acceptance.

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

APPENDIX

**Overview**

Except for the appendix part, the supplementary material presents the following sections to strengthen the main manuscript:

— Appendix A and Appendix B present the reproducibility- and the LLM usage statement.

— Appendix C presents the analysis on the weighting coefficients of our evaluation protocol.

— Appendix D presents the detailed task definitions and the prompt templates used for constructing question-answer pairs.

— Appendix E presents the concrete examples of each task under the multiple-choice format, indicating the variety of HumanVideo-MME .

## A  REPRODUCIBILITY STATEMENT

We have already elaborated on all the models or algorithms proposed, experimental configurations, and benchmarks used in the experiments in the main body or appendix of this paper. Furthermore, we declare that the entire code used in this work will be released after acceptance.

## B  THE USE OF LARGE LANGUAGE MODELS

We use large language models solely for polishing our writing, and we have conducted a careful check, taking full responsibility for all content in this work.

## C  THE CONFIGURATION OF COEFFICIENT SETTINGS FOR OEQ METRICS

To validate the robustness of our evaluation protocol, we conduct an ablation analysis on the weighting coefficients for the Open-End Question (OEQ) metrics. As summarized in Table A1, the results reveals that, although absolute scores vary under different weighting schemes, the relative performance ranking of the models remains substantially unchanged. This consistency confirms that our overall conclusions are reliable and not an artifact of a specific metric configuration.

Table A1: Impact of different OEQ metric weighting schemes on model performance and rankings. Weighting schemes are specified as tuples for $Score_F$, $Score_O$, and $Score_G$, respectively. Parenthesized numbers denote model ranks.

| Setting (Weights: $Score_F$, $Score_O$, $Score_G$) | LLaVA-Video-7B | Qwen2.5-VL-7B | Intern2.5-VL-8B | Intern2.5-VL-38B | VideoLLaMA2-7B | Qwen2.5-VL-7B | Qwen2.5-VL-32B |
|---|---|---|---|---|---|---|---|
| $Score_F$ | 0.14 | 0.15 | 0.15 | 0.17 | 0.19 | 0.22 | 0.19 |
| $Score_O$ | 0.24 | 0.33 | 0.35 | 0.37 | 0.35 | 0.47 | 0.51 |
| $Score_G$ | 0.49 | 0.56 | 0.57 | 0.53 | 0.56 | 0.64 | 0.69 |
| Proposed (0.5, 0.3, 0.5) | 0.39 (7) | 0.45 (6) | 0.46 (6) | 0.46 (4) | 0.48 (3) | 0.57 (2) | 0.59 (1) |
| Equal (0.33, 0.33, 0.33) | 0.29 (7) | 0.34 (6) | 0.35 (4) | 0.36 (4) | 0.36 (3) | 0.44 (2) | 0.46 (1) |
| Factual-Focused (0.6, 0.2, 0.2) | 0.23 (7) | 0.27 (6) | 0.28 (4) | 0.28 (4) | 0.30 (3) | 0.35 (1) | 0.35 (1) |
| Order-Focused (0.2, 0.6, 0.2) | 0.27 (7) | 0.34 (6) | 0.36 (4) | 0.36 (4) | 0.36 (4) | 0.45 (2) | 0.48 (1) |
| Semantic-Focused (0.2, 0.2, 0.6) | 0.37 (7) | 0.43 (6) | 0.43 (6) | 0.43 (6) | 0.44 (3) | 0.52 (2) | 0.55 (1) |

## D  TASK DEFINITIONS AND PROMPT TEMPLATES

To comprehensively evaluate the cognitive reasoning abilities of video MLLMs in human-centric scenes, we introduce six novel task types grounded in both perceptual attributes and higher-level inference. Each task category reflects distinct human judgment processes ranging from basic visual perception to complex mental-state modeling. Below we detail the definitions and prompt templates for constructing Question-Answer pairs associated with each category.

**Basic Attribute Recognition (BAR)** Recognizing basic attributes like age, gender, and body type supports scene understanding and provides priors for social reasoning. This task evaluates the model's ability to perceive structured information and remain robust to occlusions and variability.

The example prompts template used for constructing BAR Question-Answer pairs are as follows:

- "*Which of the following options is closest to the age range of most characters in the video?*"
- "*Based on the character's clothing/hair/makeup in the video, estimate their age.*"
- "*How many females/males are in this video?*"
- "*What is the gender of the person in the video?*"

**Face Recognition (FR)** This task focuses on recognizing well-known individuals based on facial features, supporting downstream applications such as identity grounding. This capability reflects the model's potential to link visual perception with world knowledge.

The example prompts template used for constructing FR Question-Answer pairs are as follows:

- "*Is the [character] in the video a well-known public figure?*"
- "*Is the [character] in the video a well-known celebrity?*"
- "*Is the person in the video a celebrity?*"

**Action Recognition (AR)** Human actions and postures reveal behavior, intent, and emotion. This task assesses the model's ability to understand what people are doing, how they are doing it, and with what level of engagement, supporting action description and behavior modeling.

The example prompts template used for constructing AR Question-Answer pairs are as follows:

- "*What is the main action of the people in the video?*"
- "*What is the main activity of the person in the video?*"
- "*What is the main action performed by the people in the video?*"
- "*Does character show [surprised/angry/sad, etc.] expression at [scene] in the video?*"
- "*Is the character interacting with others?*"
- "*What is the character's [head/leg/arm, etc.] posture?*"
- "*What is the main standing posture of the person in the video?*"

**Intention Inference (InI)** This task measures the model's capacity for theory of mind—inferring why someone acts a certain way. It requires integrating multimodal and contextual cues to uncover latent goals and motivations.

The example prompts template used for constructing InI Question-Answer pairs are as follows:

- "*What is the main purpose of this person?*"
- "*What is the most likely purpose of the character when interacting with others?*"
- "*What is the main goal of this character?*"
- "*What might be the motivation for the person to perform this [action]?*"
- "*The character's actions may convey which unspoken message?*"
- "*What unspoken message might the character's actions be intended to convey?*"

**Causal Reasoning (CR)** Understanding causality and temporal order is key to narrative coherence. This task evaluates the model's ability to detect causes, predict effects, and organize events chronologically—essential for structured video understanding.

The example prompts template used for constructing CR Question-Answer pairs are as follows:

- "*Which of the following sequences of actions is correctly arranged in chronological order?*"
- "*Which of the following sequences of actions is correctly ordered in time?*"
- "*What did [Character] do first, and what did they do next?*"
- "*[Character] suddenly [action]. The main impact result is?*"
- "*The most likely reason for the character suddenly [action] is?*"
- "*The most likely reason for a character suddenly displaying a certain emotion is?*"

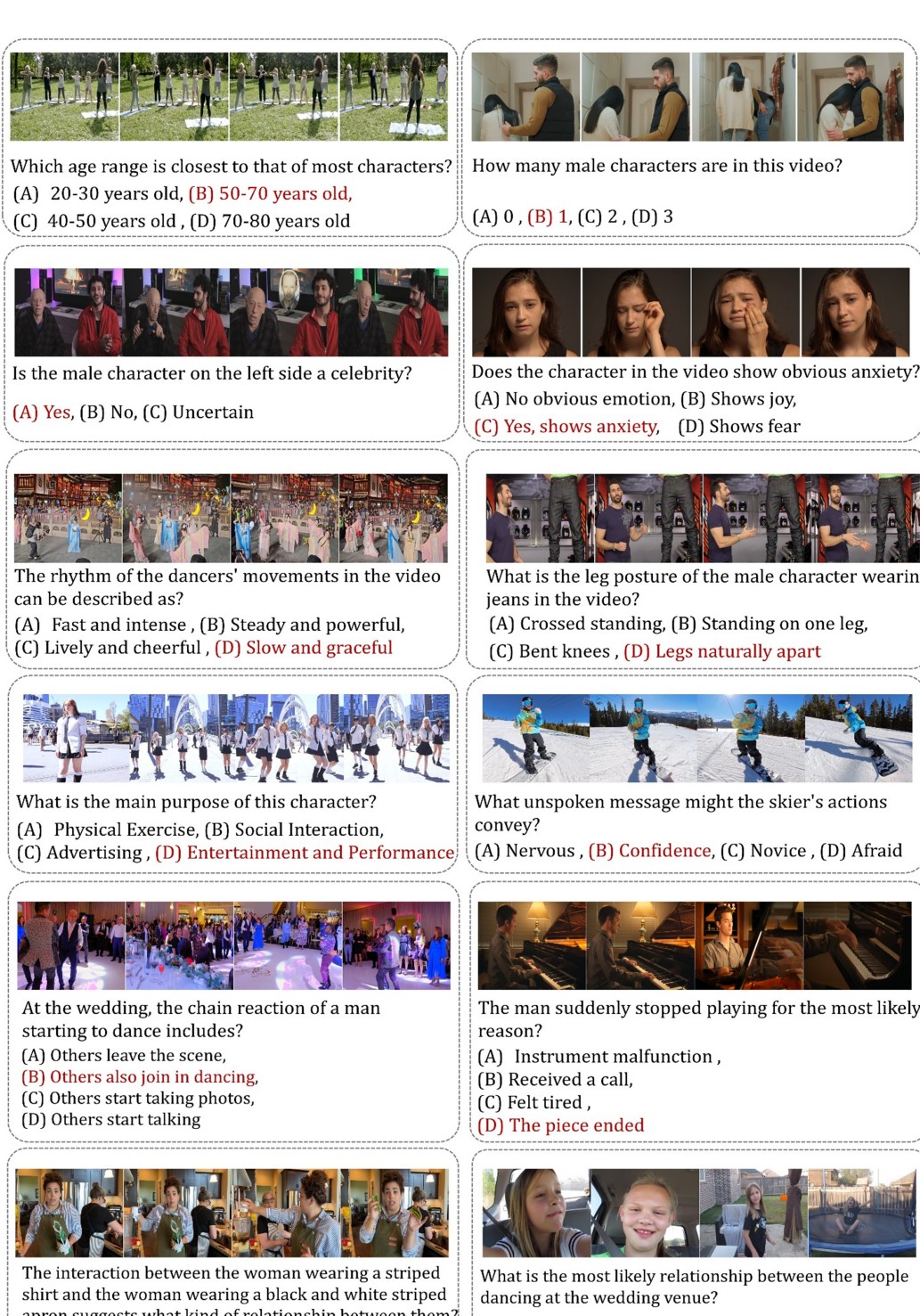

Figure A1: Concrete examples of each task of the multiple-choice format.

**Relationship Inference (RI)** Social roles and relationships shape interaction dynamics. This task tests whether the model can infer roles like leader or family member through cues such as gaze, orientation, and interaction patterns.

The example prompts template used for constructing RI Question-Answer pairs are as follows:

- "*What is the relationship between the main characters in the video?*"
- "*Do the characters in the video have a common goal?*"
- "*Is there clear leadership and subordinate relationship among the characters in the video?*"
- "*Are there obvious [cooperative/conflicts] behaviors among the characters in the video?*"
- "*The interaction between [CharacterA] and [CharacterB] suggests what kind of relationship exists between them?*"

## E EXAMPLES FOR EACH TASK

Fig. A1 shows concrete annotated examples for each task type. Each example includes a representative question, illustrating the diversity of HumanVideo-MME .

