# OpenReview forum: "HumanVideo-MME: Benchmarking MLLMs for Human-Centric Video Understanding"
_ICLR.cc/2026/Conference — Submitted to ICLR 2026_

### Official Review · Reviewer_VJrU · 2025-10-24

**Soundness:** 3
**Presentation:** 3
**Contribution:** 3
**Rating:** 4
**Confidence:** 4

**Summary:**

The authors observe a lack of benchmarks dedicated to evaluating Multimodal Large Language Models (MLLMs) in human-centered scenarios. Understanding such contexts demands not only perceptual capabilities but also advanced reasoning, presenting distinctive challenges for video understanding. To fill this gap, they propose HV-MMBench, a comprehensive benchmark encompassing 13 diverse evaluation dimensions, various temporal coverage, and extensive spatiotemporal coverage. Extensive experiments on open-source MLLMs are further conducted to evaluate their performance across these dimensions.

**Strengths:**

1. The contribution of the benchmark is clearly articulated. The authors pinpoint key challenges in human-centered multimodal video understanding, such as simplistic evaluation protocols, constrained question–answer paradigms, and limited temporal coverage. To address these issues, they introduce HV-MMBench, a comprehensive benchmark designed to overcome these limitations and provide a more robust evaluation framework for MLLMs.

2. The paper is well-structured and easy to follow, with clear and informative figures and tables that effectively support the reader’s understanding. The benchmark details are thoroughly described, encompassing the data sources, the question–answer generation process, and the distribution and categorization of the answers.

3. The authors further introduce OEQ, a hybrid evaluation framework designed to assess the causal reasoning quality of MLLMs.

**Weaknesses:**

1. To generate the ground truth answer, the authors directly apply Qwen 2.5 VL, which is similar to the process for evaluation. This raises concerns about potential model bias, as the Qwen series may have an inherent advantage; indeed, the results show that Qwen-based models outperform other series.

2. As shown in Fig. 3(e), for human basic attributes, how to capture this kind of information from the video? Is this annotation reliable? How to guarantee the correctness of the VLM-generated response.

3. While the paper identifies limitations of current MLLMs dealing with human-centric problems, it offers few novel methodological contributions to address these issues. It would be better to include some insights and methodological contributions to the paper.

**Questions:**

For certain tasks, such as face recognition, the formulation resembles a simple binary classification (e.g., determining whether a person is a celebrity or not), which is relatively straightforward. Therefore, it is important to control and monitor the difficulty levels across different task types to ensure balanced and meaningful evaluation.

---

### Official Review · Reviewer_ciHY · 2025-10-28

**Soundness:** 1
**Presentation:** 1
**Contribution:** 1
**Rating:** 0
**Confidence:** 5

**Summary:**

This paper introduces **HumanVideo-MME**, a new benchmark designed to evaluate Multimodal Large Language Models (MLLMs) on **human-centric video understanding**. The authors also propose a **novel composite metric** for evaluating causal reasoning in OEQ, combining lexical matching, structural consistency, and LLM-based semantic scoring. Several open-source MLLMs are evaluated. Results show that models perform well on **closed-form tasks** (MC/TF) but struggle significantly with **generation-based tasks** (FIB/OEQ), especially in causal reasoning.

**Strengths:**

-  **Rigorous Construction**: Uses a semi-automated pipeline with both model-generated annotations and human review for quality control.
-  **Insightful Findings**: Reveals a significant performance gap between closed-form and generative tasks, highlighting a key weakness in current MLLMs.

**Weaknesses:**

-  **Weak Justification for "Human-Centric" Focus**: The paper does not sufficiently explain why human-centric videos require a specialized benchmark beyond general video understanding.
-  **Unconvincing Metrics**: Some older/smaller models outperform newer/larger ones in certain tasks, suggesting possible flaws in metric design or benchmark construction.
-  **Limited Model Variety**: Only open-source models are tested; including proprietary models (e.g., GPT-4o, Gemini-2.5-Pro) could provide a more complete picture.
-  **Shallow Analysis**: The discussion of why models fail in generative tasks is somewhat superficial.

**Questions:**

- Why are *human-centric* videos fundamentally different from general videos? Can you provide concrete examples where general video MLLMs fail but a human-specific model would succeed?
- How did you ensure that the automated QA generation process did not introduce biases or errors that could affect benchmark reliability?
- Why do smaller models sometimes outperform larger ones? Is this due to the evaluation metric, data leakage, or other factors?
- Did you consider evaluating proprietary models like GPT or Gemini? If not, why?

---

### Official Review · Reviewer_jZjG · 2025-10-31

**Soundness:** 2
**Presentation:** 2
**Contribution:** 1
**Rating:** 2
**Confidence:** 4

**Summary:**

The benchmark shows that current MLLMs perform well on structured tasks like multiple-choice and true/false questions but struggle in generation-based tasks such as fill-in-blank and open-ended reasoning. Overall, HumanVideo-MME exposes the gap between surface pattern matching and genuine human-centric reasoning.

**Strengths:**

1. The paper introduces HumanVideo-MME, a benchmark covering 13 human-centric tasks across perception and reasoning, offering unprecedented task diversity and evaluation formats.

2. Its automated annotation pipeline combining MLLMs and human validation ensures both scalability and data quality.

3. The evaluation compares multiple MLLMs using certain metrics, revealing concrete gaps between closed-form accuracy and genuine reasoning performance.

**Weaknesses:**

1. This is a nice to know study. But i question the main research value? The dataset construction, though large-scale, heavily relies on synthetic and pre-existing public datasets, limiting novelty in raw video acquisition.

2, The evaluation design should be more balanced: open-ended reasoning tasks remain small in sample size, potentially constraining generalizability of conclusions.

3. The study focuses solely on open-source MLLMs; inclusion of closed-source or proprietary baselines (e.g., Gemini, GPT-4o) would strengthen comparative insights.

4. This preliminary dataset can be further expanded with more original, annotated human-interaction footage to increase ecological validity. The open-ended reasoning section should be enlarged with richer linguistic and contextual annotations for deeper model analysis. Incorporating cross-benchmark validation against both academic and industrial models would make the study more impactful and position the benchmark as a community-wide standard.

**Questions:**

Please see weaknessdes.

**Details Of Ethics Concerns:**

-

---

### Official Review · Reviewer_B3b7 · 2025-10-31

**Soundness:** 3
**Presentation:** 3
**Contribution:** 2
**Rating:** 4
**Confidence:** 4

**Summary:**

This paper proposes a human-centric video understanding benchmark, aiming to evaluate models' capabilities in comprehending video content from a human-oriented perspective. The benchmark incorporates four question types: multiple-choice questions (MCQ), fill-in-the-blank, true/false judgment, and open-ended questions. To assess model performance across these different question formats, the paper reports corresponding metrics, with results from models like Qwen-2.5-VL series provided as experimental demonstrations.

**Strengths:**

The focus on a "human-centric" video understanding benchmark fills a potential gap in existing evaluations that may overly emphasize task-specific or non-human-perspective video analysis. Shifting the focus to human-centric comprehension introduces a novel angle for assessing video understanding models, which aligns with real-world scenarios where human relevance is crucial.

**Weaknesses:**

- Redundant question type design: Incorporating four question types (selection, fill-in-the-blank, judgment, open-ended) within a single benchmark is unnecessary. A benchmark's effectiveness lies in its ability to accurately and efficiently measure the target capability (human-centric video understanding). If one or two question types are most suitable for this purpose, prioritizing those would reduce complexity and focus the evaluation. The current multiplicity may dilute the benchmark's core value and increase the burden of model adaptation without proportional gains in assessment accuracy.
- Confusing metric & difficulty levels across question types: The performance metrics of different question types show implausible discrepancies. Taking Qwen-2.5-VL-32B in action recognition as an example: MCQ (4 options) achieves 96.9% accuracy, while true/false (2 options) only reaches 83.1%, and fill-in-the-blank precision@1 is as low as 17.2%. This raises critical concerns: (1) For MCQ, the significantly higher accuracy than true/false suggests that distractors may be overly simple, failing to effectively distinguish model capabilities. (2) The extreme gap between fill-in-the-blank (17.2%) and the other two closed-ended question types indicates potential flaws in fill-in-the-blank question design (e.g., ambiguous answer standards, overly narrow answer scopes) or metric calculation, making cross-question-type comparisons meaningless.
- Potential bias in question generation and model evaluation: Using Qwen-2.5-VL-72B for question generation may limit the benchmark's difficulty to the model's own capabilities, resulting in a ceiling effect. Moreover, evaluating Qwen series models on a benchmark created by the same model family could introduce inherent bias, as the models may be more familiar with the question style or reasoning patterns employed during generation. This undermines the benchmark's ability to objectively assess general video understanding capabilities across diverse model architectures.

**Questions:**

Please issue the concerns in the weakness section.

**Details Of Ethics Concerns:**

The benchmark undoubtedly involve significant number of human subjects with clear identity features (see Figure A1). However, the authors did not include further discussion and directly claim "no direct involvement of human subjects, and no foreseeable risk of harm" in the ethics statement. I'm not sure whether this is appropriate.

---

### Meta-Review · Area_Chair_yQTM · 2026-01-06

**Summary:**

The authors identify a lack of benchmarks for evaluating MLLMs in human-centered scenarios and propose a human-centric video understanding benchmark. Reviewers generally agree that the work addresses a potential gap in existing evaluation practices. However, multiple reviewers raise four recurring concerns. First, the justification for focusing on “human-centric” videos is insufficient, as the need for a dedicated benchmark beyond general video understanding is not clearly articulated (ciHY). Second, the use of Qwen-2.5-VL-72B for both question generation and model evaluation introduces potential bias (B3b7, VJrU). Third, the evaluation is limited to open-source MLLMs, lacking comparisons with proprietary models (jZjG, ciHY). Finally, some experimental results are confusing, such as cases where smaller models outperform larger ones without adequate explanation (B3b7, ciHY). Overall, these issues suggest that substantial revision is required to strengthen the paper.

**Reviewer Concerns:**

There was no author response, so no concerns were addressed.

**Reviewer Scores:**

There was no author response, so the reviewers would not change their scores.

---

### Decision · Program_Chairs · 2026-01-26

Reject